# Rare Non-Neuroendocrine Pancreatic Tumours

**DOI:** 10.3390/cancers15082216

**Published:** 2023-04-09

**Authors:** Agata Mormul, Emilia Włoszek, Julia Nowoszewska, Marta Fudalej, Michał Budzik, Anna Badowska-Kozakiewicz, Andrzej Deptała

**Affiliations:** 1Students’ Scientific Organization of Cancer Cell Biology, Department of Oncology Propaedeutics, Medical University of Warsaw, 01-445 Warsaw, Poland; 2Department of Oncology Propaedeutics, Medical University of Warsaw, 01-445 Warsaw, Poland

**Keywords:** oncology, pancreatic tumour, intraductal papillary mucinous neoplasm, mucinous cystadenoma, serous cystic neoplasm, solid pseudopapillary neoplasm, acinar cell carcinoma, pancreatoblastoma

## Abstract

**Simple Summary:**

Pancreatic ductal adenocarcinoma accounts for 85% of non-neuroendocrine pancreatic lesions. The remaining 15% consists of numerous diverse neoplasms, both malignant and benign. We gathered the latest data about the epidemiology, diagnosis, biomarkers and management of six rare pancreatic tumours: intraductal papillary mucinous neoplasm, mucinous cystadenoma, serous cystic neoplasm, acinar cell carcinoma, solid pseudopapillary neoplasm and pancreatoblastoma. Frequent guideline updates can help to avoid misdiagnosis, which could lead to unnecessary resections or oversight of malignant transformations.

**Abstract:**

The most common tumour of the pancreas is ductal adenocarcinoma (PDAC). It remains one of the most lethal non-neuroendocrine solid tumours despite the use of a multi-approach strategy. Other, less-common neoplasms, which are responsible for 15% of pancreatic lesions, differ in treatment and prognosis. Due to the low incidence rate, there is a lack of information about the rarest pancreatic tumours. In this review, we described six rare pancreatic tumours: intraductal papillary mucinous neoplasm (IPMN), mucinous cystadenoma (MCN), serous cystic neoplasm (SCN), acinar cell carcinoma (ACC), solid pseudopapillary neoplasm (SPN) and pancreatoblastoma (PB). We distinguished their epidemiology, clinical and gross features, covered the newest reports about courses of treatment and systematised differential diagnoses. Although the most common pancreatic tumour, PDAC, has the highest malignant potential, it is still essential to properly classify and differentiate less-common lesions. It is vital to continue the search for new biomarkers, genetic mutations and the development of more specific biochemical tests for determining malignancy in rare pancreatic neoplasms.

## 1. Introduction

Pancreatic ductal adenocarcinoma (PDAC) is the most common epithelial tumour of the pancreas, accounting for roughly 85% of non-neuroendocrine pancreatic lesions. PDAC is generally associated with poor prognosis and an aggressive course [1]. Furthermore, it is one of the most lethal solid tumours despite the use of a multi-approach strategy. The remaining 15% consists of numerous diverse neoplasms, both malignant and benign [1,2]. It is important to mention that some rare pancreatic tumours, such as mucinous cystic neoplasm (MCN) and intraductal papillary mucinous neoplasm (IPMN), can transform into PDAC [1,2]. Due to the heterogeneity of these rare tumours, accurate diagnosis is crucial for choosing the appropriate treatment, as management varies in different types. Incidental discoveries still represent a large percentage of newly diagnosed patients. For example, up to 15% of pancreatic cystic lesions (PCLs) are detected by MRIs performed for unrelated reasons [3]. However, more precise methods for diagnosis and malignancy markers still need to be developed to correctly identify lesions, implement appropriate treatment and avoid unnecessary surgery [2]. A low incidence rate of lesions other than PDAC in the pancreas results in fewer articles and less data; hence, in this review, we compiled the latest information about selected rare, non-neuroendocrine pancreatic tumours involving their clinical features, differential diagnoses, prognoses and treatments. The aim of the study is to propagate knowledge about rare pancreatic tumours to improve the precision of their diagnoses.

## 2. Classification

Below, we present the 2019 WHO pathomorphological classification of pancreatic tumours featured in this article, with their corresponding ICD-O codes, based on histologic, but not molecular appearance (Table 1) [2]. These lesions can as well be categorised into cystic and solid, which is commonly used in literature and has been used to organise this article [1]. Most common mutations associated with benign and malignant pancreatic tumours are presented in Figure 1.

## 3. IPMN

### 3.1. Epidemiology, Clinical and Gross Features

Intraductal papillary mucinous neoplasm (IPMN) is an epithelial cystic neoplasm of the pancreas characterised by the formation of papillae inside the pancreatic duct and the production of mucin. IPMNs can be classified according to their main location into main duct (MD) type (16–36%), branch duct (BD) type (40–65%) and mixed, involving both main duct and side branches (15–23%) [5]. Another classification can be based on cytological features and mucin immunochemistry, dividing IPMNs into three types: gastric, intestinal, and pancreato-biliary (Table 2). Previous classification included an oncocytic subtype; however, it has been recognised as a distinct type named intraductal oncocytic papillary neoplasm (IOPN) [2].

Typically, IPMNs progress from low-grade to high-grade dysplasia, and those neoplasms with high-grade dysplasia may develop into invasive carcinomas [5]. Although most IPMNs are benign tumours, there is a 50% risk of malignant transformation in MD-IPMN and mixed types, while in BD-IPMN that risk is at 15% (Table 3) [6]. More recent, although retrospective, studies have reported malignancy rates of MD-IPMN and BD-IPMN to be 68% (66.0–70.9%) and 48.6%, respectively [7]. Survival analysis conducted by Valsangkar et al. (2012) for MD-IPMN and BD-IPMN showed the 5-year survival rates to be 83% and 88% (Table 3), while the 10-year survival rates were 78% and 80%, respectively [4]. In invasive IPMN, the observed rates were significantly lower: the 5-year survival rate could be as low as 16.6% (median overall survival of 11 months) [8]. Other studies observed overall survival for invasive IPMN at 17–76.6 months [9,10,11]. Noteworthily, IPMNs constitute approximately five percent of precursors of pancreatic ductal adenocarcinoma (PDAC) [4,5].

IPMNs constitute 38% of pancreatic cystic neoplasms and 3–5% of all pancreatic neoplasms [1,12]. It is usually diagnosed in the seventh decade with no significant predilection towards any sex (Table 3) [5,12]. The main location is in the head of the pancreas (70%), while about 20% are located in the body or tail and 5–10% of the lesions are diffused [5]. 

More than 50% of patients present with symptoms, mostly due to the papillary growth of ductal epithelium or duct obstruction with mucin, which may result in acute pancreatitis. Other symptoms include abdominal pain, jaundice, diarrhoea and weight loss [6].

### 3.2. Diagnosis

According to European evidence-based guidelines, the accuracy for identifying the specific type of PCN with MRI is 40–95%, while the accuracy for CT is 40–81% [13]. For determining the relation to the pancreatic duct, contrast-enhanced MRI is preferred. In cases of diffused dilution of the duct, pancreatoscopy may help assess the scale of the disease [2]. EUS is recommended to identify whether the neoplasm is mucinous. Combining EUS with fine needle aspiration (FNA) helps to differentiate malignant or invasive forms from benign lesions and further identify the malignancy [2,13]. An analysis of specific biomarkers may provide crucial information for the management of the lesion, as well as help to discriminate between different types of pancreatic tumours [14]. New diagnostic technologies are currently emerging, including mutational analysis of cyst fluid, needle-based confocal laser endomicroscopy (nCLE), micro-forceps biopsy, single operator pancreatoscopy and contrast-harmonic enhanced endoscopic ultrasound (CH-EUS) [15].

Carcinoembryonic antigen (CEA), a glycoprotein associated with cell adhesion, is used as a biomarker for various gastrointestinal malignancies. It has proven useful in differentiating malignant lesions from benign ones, as well as in IPMN diagnosis; however, it does not aid in distinguishing IPMNs from mucinous cystic neoplasms (MCNs), as an elevated level of CEA in cystic fluid is present in both lesions (Table 4) [14]. Nonetheless, a CEA level > 192 ng/mL may differentiate IPMNs and MCNs from other pancreatic cysts. Additionally, combining levels of CEA > 5 µg/L and CA19.9 > 37 U/mL showed even higher sensitivity and specificity in determining malignancy in IPMNs (Table 5) [14].

For determining connection to the pancreatic duct, thus differentiating IPMNs and pseudocysts from MCNs and serous cystic neoplasms (SCNs), amylase has proven useful. Furthermore, combining high amylase levels (>8500 U/L) with low CEA levels (<30 ng/mL) helps identify pseudocysts and differentiate them from IPMNs, which contain high levels of CEA, as mentioned above (Table 5) [1]. High amylase, together with lipase, was also found to be correlated with the malignancy of IPMN in patients without pancreatitis [14].

For differentiating high-risk and malignant IPMNs from low-risk neoplasms, monoclonal antibody (mAb) Das-1, IMP3, and various DNA mutations may be used. mAb Das-1, which acts against colonic epithelium, has proven useful in determining malignancy, with 89% sensitivity and 100% specificity in a retrospective study of 27 patients by Das et al. (2014) [21]. These results have been validated by the same group in a larger patient cohort, using 169 cyst fluid samples, and the sensitivity and specificity of detecting high-risk IPMNs were 88% and 99%, respectively [16]. 

KRAS mutations are found in all types of IPMN and are thought to be an early event in the pathogenesis of the lesion. The alterations of KRAS found in cystic fluid have proven to be highly specific (92–96%) in diagnosing mucinous cysts; however, their sensitivity was quite low (33–45%). Specificity and sensitivity of mucinous differentiation may be increased by the conjunction of KRAS and GNAS mutations to 100% and 65%, respectively (Table 5) [14]. A recent study has also described the RNF43 mutation, which leads to deficiency of Rnf43, as a significant predictor of malignant transformation, as it potentiates the hyperactivity of KRAS [22]. 

There have been multiple studies analysing resected tissue to determine miRNA profiles of low- and high-risk IPMNs, as well as PDAC [14]. Four biomarkers (miR-21-5p, miR--483-3p, miR-708-5p and miR-375) were observed to differentiate between IPMN and PDAC with a 95% sensitivity and 85% specificity. Furthermore, a study by Vila-Navarro et al. (2017) has found that miR-93 helped to discriminate PDAC and IPMN patients from healthy controls with 100%/96% sensitivity and 89%/88% specificity [23]. miRNA may be highly useful in predicting malignant transformation; however, it has a limitation due to its low yield from pancreatic fluid samples [14].

### 3.3. Management

The risk of developing PDAC in MD-IPMN and mixed IPMN is significantly high; thus, patients with resectable lesions should undergo resection [13]. Main pancreatic duct dilatation ≥ 10 mm, enhancing mural nodes ≥ 5 mm, positive cytology for malignant or high-grade dysplasia, the presence of solid mass and jaundice are absolute indications for surgical resection. Relative indications include growth rate ≥ 5 mm/year, levels of serum CA19.9 ≥ 37 U/mL, pancreatic duct dilatation of 5–9.9 mm, cyst diameter ≥ 40 mm, enhancing mural nodules < 5 mm, acute pancreatitis caused by IPMN and new onset of diabetes mellitus [13]. Respective indications are applied to patients with BD-IPMN, which has a lower risk of malignancy; however, since it often presents multifocally, there is a potential for developing PDAC [13,24].

Adjuvant treatment may be beneficial (both radiological and chemotherapeutic) in cases of the invasive component (genuine carcinoma) in IPMN. Current schemes include FOLFIRINOX (folinic acid, 5-FU (fluorouracil), irinotecan and oxaliplatin) for younger patients with better performance status (PS), as well as gemcitabine, used in older patients or patients with worse PS (Table 6) [25]. However, it is important to note that current data are variable since they are mostly based on small observational studies. Most benefits were observed in patients with node-positive, higher-stage tumours [26,27,28]. Interestingly, in some studies it was observed that adjuvant treatment in patients with low grade invasive IPMN did not have a positive effect compared to surgery alone [26]. The European guidelines recommend adjuvant chemotherapy in all IPMN-associated invasive carcinomas [13].

Patients who underwent IPMN resection should be under lifelong surveillance. For high-grade dysplasia IPMN and MD-IPMN, a close follow-up every 6 months for the first 2 years is recommended, followed by yearly surveillance, which should include clinical evaluation, serum CA19.9 tests and MRI and/or EUS. Recurrence of the lesion is possible for up to 10 years after resection [13].

When the lesion is resectable, but there are no indications for urgent surgery (cyst lacking high-risk features, patients with co-morbidities), guidelines recommend a follow-up every 6 months for the first year from diagnosis, and after that once a year, until the patient no longer fit for surgery or some indications for such a procedure arise [13]. Nevertheless, the American Gastroenterological Association recommends surveillance for every cyst without worrisome features: for cysts < 1 cm, CT/MRI after 6 months, then every 2 years if no change; for cysts whose size is 1–2 cm, CT/MRI every 6 months during the first year, then once a year for 2 years, then every 2 years if nothing changes; for cysts whose size is 2–3 cm, EUS in 6 months, then once a year, alternating between EUS and MRI as appropriate; for cysts whose size is >3 cm, resection is strongly recommended for young, fit patients, otherwise close surveillance (alternating EUS with MRI) every 3–6 months [24].

## 4. MCN

### 4.1. Epidemiology, Clinical and Gross Features

Mucinous cystadenoma (MCN) is one of the pancreatic cystic neoplasms (PCN). It is an epithelial tumour that produces mucin and forms cysts in the pancreas. They account for about half of PCNs, with the others being serous cystadenoma (SCN) and IPMN [37]. MCNs account for approximately 1% to 2% of all pancreatic tumours and represent 2–5% of all exocrine pancreatic tumours [14]. The mean age at diagnosis is 48 years (range 4–95 years). These lesions are predominantly found in women of perimenopausal age (F:M > 20:1) (Table 3) [37]. The great majority are found in the body and tail, forming a singular, well-encapsulated, frequently unilocular mass. Although they are slow-growing, MCNs are usually large (over 4 cm) and can grow up to 25 cm (Table 3). Abdominal pain, recurrent pancreatitis, gastric outlet obstruction, jaundice, palpable mass in the upper abdomen and weight loss are all symptoms of MCNs. However, MCNs are often discovered incidentally [37,38].

### 4.2. Malignancy

MCNs exhibit a spectrum of neoplastic transformation from benign to borderline or malignant. Several studies have attempted to find a marker that can distinguish between these three. The risk of malignancy increases if the lesion is multilocular, contains mural nodules, papillary projections or thick and irregular walls and peripheral calcifications [9,24]. MCNs that show no signs of malignancy are classified as premalignant. It is estimated that invasive carcinoma represents between 4.4 and 16.6% of all MCNs (Table 3) [39].

### 4.3. Diagnosis

To classify a mucin-lined cystic lesion as MCN, two conditions must be fulfilled. Firstly, MCNs do not communicate with the pancreatic ductal system, which is the key difference that distinguishes them from IPMNs [2]. Secondly, the presence of ovarian-type stroma is the pathognomonic finding in a mucinous cystic neoplasm [13].

It is still unclear whether the resemblance between MCN’s stroma and ovarian stroma is purely morphologic or also shares more functional similarities. A number of tissues and tumours, including the ovarian-type stroma of MCN, have been shown to express oestrogen receptors (ER) and progesterone receptors (PR). On the other hand, inhibin, which has been shown to be expressed in pancreatic MCNs, has a very limited expression, confined to ovarian sex cord stromal components and placental cells. This finding further supports its similarity to true ovarian stromal tissue and may imply that complex hormonal interactions are involved in the pathogenesis. However, it is still unknown whether this expression is limited to pancreatic MCNs and whether it serves any diagnostic purpose, especially in needle biopsies of pancreatic lesions [40].

There are numerous hypotheses explaining the possible relationship between ovarian-type stroma and MCN pathogenesis. These tumours may be the result of müllerian remnants that were misplaced during embryogenesis. The most likely theory is that because of hypersensitization to female sex hormone stimulation, endodermally derived epithelium and primordial mesenchyme in the pancreas may begin proliferating. The ovarian-type stroma seen in MCNs has never been documented to contain follicles, making the second theory—that MCNs may originate from an ectopic ovary incorporated in the pancreas—less likely. The third theory is that the expression of ER, PR and α-inhibin might simply be a secondary phenomenon unrelated to tumorigenesis [41].

A combination of the clinical and imaging characteristics provides the best initial preoperative diagnosis of the cyst type. The preferred methods of imaging are MRI or contrast-enhanced CT, which are the best for assessing pancreatic ductal communication. The diagnostic investigation should also include EUS with cyst fluid aspiration [37]. Research has shown increased sensitivity when combining MRI with EUS in comparison to using only one of these methods when it comes to the identification of not just MCNs, but also other pancreatic cancers and cysts with high-grade dysplasia [14]. PET-CT combined with CT was also shown to be more effective than CT alone in differentiating benign from malignant lesions. If mural nodules are present in the neoplasm lesion, the sensitivity and specificity of a CT scan in proving malignancy are around 100% and 98% [14]. A benign MCN on MRI usually presents as a mass with thin non-enhancing walls with no mural nodules. However, a heterogenous signal on T2-weighted MRI, thick walls (≥5 mm), mural nodules (≥9 mm) or enhancing septa can suggest malignancy [42].

After resection, there is a possibility of histological identification of the ovarian stroma. Oval nuclei with spindle cell neoplasm that stain positive for oestrogen receptors distinguish these stromal cells. Luteinized cells with a transparent cytoplasm that stain positive for inhibin can also occasionally be observed in the cysts [2].

### 4.4. Differential Diagnosis

In many cases, MCNs must be differentiated from other pancreatic cystic neoplasms, especially IPMNs. In addition, the differentiation should include pseudocyst, oligocystic serous cystadenoma, and cystic PNETs (pancreatic neuroendocrine tumours). Imaging studies and cyst fluid sampling, in association with clinical features, are mandatory for a correct diagnosis [1].

Pancreatitis is seen in nine percent of MCN patients and is associated with pancreatic duct compression. Pancreatitis presented along with a big, well-circumscribed cyst might be diagnosed as a pseudocyst associated with chronic pancreatitis or MCN and requires additional imaging and invasive procedures [14]. 

### 4.5. Biomarkers

The primary method for diagnosing MCN is the analysis of the fluid aspired from the cyst. Elevated CEA is a marker that distinguishes mucinous from non-mucinous cysts, but does not correlate with the presence of malignancy [43]. For the identification of a mucinous cyst, a cut-off of 192–200 ng/mL is accurate in about 80% of patients (with high specificity but low sensitivity) [44]. A viscosity test can also be performed on cystic fluid. The median string sign of mucinous cysts is 3.5 mm, in comparison to the mean string sign of 0 mm in non-mucinous cysts [2]. Chemical analyses of fluid CEA level and the viscosity test can be useful, but will not distinguish MCNs and IPMNs (Table 4). Due to no connection with the pancreatic duct, MCNs are presented with a lower amylase level (<250 U/L), contrary to IPMNs [2].

Other biomarkers, such as mucins (MUC1, MUC2, MUC5AC), CA19.9 and prostaglandin E2 (PGE2) can help identify MCNs, but more studies are needed to determine the sensitivity, specificity and utility of these biomarkers in the clinical setting (Table 5) [14].

### 4.6. Metabolomics

Metabolomics, especially glucose and kynurenine, can also be helpful in the diagnostic process. MCNs have significantly lower glucose and kynurenine levels than non-MCNs. Glucose sensitivity and specificity for differentiating MCNs are 94% and 64% (<66 mg/mL) (Table 5) [14]. Other studies show that if the glucose level of the cystic fluid is below 50 mg/dL, the sensitivity is higher than 90% with a specificity of 87% [14]. Kynurenine sensitivity/specificity was around 90%/100% [14]. Metabolomics are promising in the prediction of cystic lesion stratification, but seem not to be useful in assessing malignancy.

### 4.7. Genetics

The use of molecular analysis to diagnose cyst fluid is still in its early stages. The most common mutations in MCNs are the KRAS (Kirsten rat sarcoma virus), TP53 (Tumour protein P53) and RNF43 (Ring Finger Protein 43) mutations [17]. KRAS mutations help to confirm the diagnosis of a mucinous cyst that is not necessarily malignant. The role of GNAS mutations in distinguishing between significant mucinous cysts and indolent cysts that may be treated conservatively is also being investigated [44]. It is important to note that KRAS/GNAS combined mutations have lower specificity for MCNs in comparison to IPMN [2]. A study from 2011 stated that a molecular analysis for GNAS mutations can distinguish MCN from BD-IPMN [43]. In IPMN and MCN diagnosis, adding a CEA test to the KRAS/GNAS analysis enhances sensitivity and accuracy [17]. Recent studies suggest that SMAD4 and TP53 alterations may be associated with invasive or high-grade lesions [37].

### 4.8. Management

Unfortunately, there is no precise distinction between invasive and non-invasive MCNs. Therefore, the risk of the lesion becoming malignant should be weighed against the operational risk. 

MCNs with a diameter of more than 40 mm should be surgically removed, according to European guidelines (Table 6). Regardless of the size, resection is suggested for MCNs that are symptomatic or have risk factors (for example, a mural nodule) [13]. It is important to consider how quickly an MCN’s size expands. Some case studies suggest faster MCN growth during pregnancy, possibly leading to tumour rupture. As a result, women with MCNs should be properly monitored during pregnancy [45].

Some studies suggest that surgical resection is recommended for all patients. MCNs are most commonly seen in the pancreatic body and tail. That is why standard oncologic resection, mostly involving distal pancreatico-splenectomy with or without lymph node dissection, is usually required [37]. However, in the absence of risk factors and in cases of MCNs smaller than 4 cm, other surgical procedures should be considered (parenchyma-sparing resections, such as middle pancreatectomy, distal pancreatectomy with spleen preservation, and laparoscopic procedures) [37]. The results of a 2020 study of 707 patients with MCNs show improved survival in patients undergoing surgical intervention. Patients who undergo pancreatectomy have a higher survival rate, especially if MCN coexists with carcinoma in situ. It is also important to underline that 75% of patients who had resections had already developed invasive adenocarcinoma. Nodal status is important in assessing future prognosis. This study also suggests that the presence of invasive disease is associated with advanced age, but not cyst size [46].

Other recent studies favour surveillance (consisting of MRI and/or EUS) over surgical resection in selected cases of patients with MCNs without malignant features, primarily because of the high risk of misdiagnosis and low risk of malignancy [39]. It is estimated that less than 0.4% of MCNs ≤ 3 cm without dubious features are considered invasive [38]. There is a 20% chance of misdiagnosis for pancreatic cysts first considered to be MCNs based on clinical and radiological findings. The routine use of EUS can minimize the number of mistakes made during the diagnostic process. That is why a thorough diagnostic assessment must be carried out before deciding on a surgical approach [39].

Postoperative complications, such as pancreatic fistulae requiring prolonged drainage or haemorrhage, should also be considered. After distal pancreatectomy, up to 20%, and after pancreaticoduodenectomy, up to 10% of patients may develop new-onset diabetes mellitus [37].

In 2018 European evidence-based guidelines on pancreatic cystic neoplasms, the updated adjuvant treatment recommendations for MCN-associated invasive carcinoma are similar to those for sporadic pancreatic adenocarcinoma. However, it is important to point out that there is not enough evidence to either prove or disprove this strategy. Given the significant heterogeneity among studies, no precise advice can be made on the type of chemotherapeutic agent that should be used. The most commonly used medications are gemcitabine-based and 5-fluorouracil-based, which are also used as adjuvant therapy for pancreatic adenocarcinoma (Table 6). There are not enough data to provide recommendations for postoperative adjuvant treatment for malignant MCNs. Palliative chemotherapy may be considered for patients with non-resectable, recurrent or metastatic malignant components of MCN [9].

Cyst ablation with ethanol or paclitaxel injection and radiofrequency ablation are not standard treatments. Several studies analysed if ethanol, either alone or in conjunction with paclitaxel, may be used to ablate pancreatic cyst epithelium to avoid surgery. The outcomes were diversified, with cyst clearance reported in 33–79% of cases [47]. However, it is not proved that decreasing MCN size reduces the risk of progression to high-grade dysplasia or pancreatic cancer.

## 5. SCN

### 5.1. Epidemiology, Clinical and Gross Features

Serous cystic neoplasm is a benign tumour, which constitutes 11–16% of all cystic pancreatic lesions and 1–2% of all pancreatic tumours [1,12,29]. It mostly affects women, with a female-to-male ratio of 2:1, in the sixth and seventh decades (Table 3), and can be also associated with von Hippel-Lindau disease [12,18,48,49]. The lesion presents as solitary, well-circumscribed cysts filled with serous fluid, with a lining consisting of cuboidal or flattened epithelium (Table 3) [48]. The serous fluid, which is rich in glycogen, stains positive with periodic acid-Schiff (PAS) stain [1]. SCN can be divided into four morphological types: microcystic, most typical, consisting of multiple cysts < 2 cm with thin fibrous septa and sometimes a honeycomb appearance; macrocystic, with cysts ≥ 2 cm; mixed microcystic-macrocystic; and solid. Cystic calcifications can occur [49]. SCN is a non-mucinous pancreatic cyst, which differentiates it from MCN and IPMN [48]. Another significant difference between these tumours is that SCN has a minimal malignant potential; in a study conducted by Jais et al., among 2622 patients, only three cases (0.1%) of serous cystadenocarcinomas were reported [49]. Two of them presented with liver metastasis, and one with distant hepatic artery lymph nodes metastases; however, in a recent review, Walsh et al. (2021) point out that some of these cases would not fulfil current WHO criteria for malignancy, and therefore the existence of serous cystadenocarcinoma is dubious [2]. This is particularly significant, since an inaccurate or uncertain diagnosis may lead to unnecessary surgery, in which case the mortality of patients could increase; the previously mentioned study reported disease-specific mortality as 0.1%, while operative mortality was 0.6% [49]. Due to its atypical morphology, especially concerning unilocular and macrocystic SCN, it has been frequently misdiagnosed as other cystic lesions. In 27 cases studied by Amico et al., 10 patients underwent surgery based on suspicion of MCN (seven cases), IPMN (two cases) and cancer (one case) [50].

Serous cystic neoplasms are usually asymptomatic (61%) and are mostly detected incidentally. Patients presenting with symptoms have mostly non-specific abdominal pain (27%), pancreaticobiliary symptoms (9%), diabetes mellitus (5%) and other symptoms (4%; abdominal mass, asthenia, nausea and vomiting) [49]. 

### 5.2. Diagnosis

Diagnosis of pancreatic cysts is usually based on radiological imaging; however, in the case of SCN, it can lead to misdiagnosing the lesion as MCN or neuroendocrine tumours, especially since it can be accompanied by the latter type in up to 15% of cases [2,13]. Accurate diagnosis of SCN is particularly important since it is a benign lesion, and the management differs significantly from the management of MCN or IPMN, which have greater malignant potential and which early stage SCN can mimic [18]. 

Fluid CEA levels, as mentioned before in the diagnosis of IPMNs, have proven useful for differentiating MCN and IPMN from SCN, with a 73% sensitivity and 84% specificity (Table 4) [14]. Furthermore, a different study has shown that a threshold of ≤10 ng/mL can identify SCN with 95.5% sensitivity and 81.5% specificity [18]. 

Vascular endothelial growth factor-A (VEGF-A) levels were shown to be elevated in SCNs in comparison to pseudocysts, MCNs, IPMNs and SPNs in a study of 149 patients’ fluid samples. The sensitivity and specificity of VEGF-A alone were 100% and 83.7%, respectively, while combining VEGF-A with CEA caused an increase in specificity to 100%, with a sensitivity of 95.5% [18]. This makes for a promising specific test. Elevated levels of VEGF-A occur in neuroendocrine tumours (NETs) as well, however, the latter usually do not form cysts [19].

In a recent study conducted by Wong et al. (2021), the authors compared four potential immunomarkers as distinguishing ones between SCN and NETs: GLUT1, VEGF-A, MUC6 and calponin (Table 5). GLUT1 and VEGF-A were shown to be suboptimal due to their expression in 10% and 44% of NETs, respectively. However, cytoplasmic expression of MUC6 proved 100% specific and sensitive as an SCN marker. Calponin, compared to inhibin, showed less sensitivity, but more specificity in diagnosing SCNs. The two biomarkers—MUC6 and calponin—may be useful as complementary studies to inhibin [19].

Principal component analysis (PCA) may also be used to differentiate IPMNs and SCNs. IPMN seems to involve many more pathways associated with lipid metabolism than SCN, which are phosphatidylethanolamine synthesis, phosphatidylcholine synthesis, taurine metabolism, oxidation of branch-chain fatty acids and sphingolipid metabolism. In a study by Gaiser et al., PCA of cystic fluid or plasma helped to discriminate between IPMN and SCN with 100% accuracy [14].

### 5.3. Management

According to European evidence-based guidelines, SCN mortality is nearly zero and asymptomatic patients should be initially followed up for 1 year [13]. Meanwhile, American College of Radiology (ACR) recommendations do not suggest any initial follow-up and recommend follow-up based only on symptoms [24]. 

Due to the high rates of major complications of pancreaticoduodenectomies in benign cases, specific factors should be taken into consideration before qualifying the patient for surgery (Table 6) [28]. Such a procedure is recommended only for symptomatic patients by both the ACR and European committee; however, the ACR suggests a cut-off for cysts of >4 cm, whereas the European guidelines are based solely on symptoms related to the compression of adjacent organs [13,24]. The size of about 40% of the cysts increases; however, the growth rate is slow, and onset of new symptoms is rather rare [13].

## 6. SPN

### 6.1. Epidemiology, Clinical and Gross Features

Solid pseudopapillary neoplasm (SPN) is also known as solid pseudopapillary tumour (SPT), solid-cystic tumour, solid pseudopapillary epithelial neoplasm (SPEN) and Frantz tumour. SPN accounts for approximately two percent of exocrine pancreatic tumours [32]. It is low-grade malignant with a predisposition to present in young females, with a peak incidence at 28 years of age (Table 3). Specific symptoms or endocrine disorders are not present, and additional indicators of a tumour are within the normal range [1]. In some cases, patients might present with a gradually enlarging abdominal mass or generalised abdominal pain caused by compression to the pancreas, but in about 15%, they are asymptomatic [30]. Some patients had a history of pancreatic trauma or were diagnosed with elevated CA 19-9 levels [32]. As is true for most of the pancreatic tumours, SPN in the early stage is often found incidentally. Tumours are located most frequently in the tail of the pancreas (Table 3). It is crucial to distinguish SPN from other pancreatic neoplasms, because in most cases complete resection of this tumour is a curative treatment with a good prognosis [30].

### 6.2. Diagnosis

SPN presents vague radiologic features; nevertheless, its histologic features are usually specific [1]. The architecture of tumour cells is pseudopapillary, with nuclear staining by β-catenin, synaptophysin, chromogranin and membranous presentation of CD10 in almost all specimens, which helps to distinguish SPN from neuroendocrine neoplasms (Table 5). Molecular investigations of SPN show its differences from ductal adenocarcinoma, since it lacks the KRAS, TP53 and SMAD4 mutations. Additionally, in SPN, CTNNB1 mutations almost always occur (Table 3) [30].

Imaging studies used in differential diagnosis are ultrasonography, magnetic resonance imaging (T1- and T2-weighted images) and computer tomography (where combined solid, cystic, haemorrhagic, calcified and necrotic components can be seen) [1]. A classic X-ray will only show possible calcification, and therefore it is not a standard diagnostic method [30].

### 6.3. Management

SPNs are low-grade tumours with a good prognosis; nevertheless, they are treated with surgical resection (Table 4). Due to its low malignancy, it is uncommon to find metastases in lymph nodes; however, searching for positive lymph nodes should be considered in SPN which presents atypical cellular features, elevated proliferative index or extensive necrosis [30]. Margin-negative surgical resection is curative in most patients, and the most common operations are distal resection of the pancreas or the Whipple procedure [32]. About 15% of patients demonstrate malignant transformation of SPN and features demonstrating invasion of adjacent organs [30]. Despite the delayed diagnosis, the overall prognosis of these tumours remains good even with local recurrence and metastasis treated with an aggressive surgical approach [33]. The role of adjuvant or neoadjuvant chemotherapy and radiotherapy is not yet well defined, due to its high rate of resectability limiting the need for adjuvant therapy. There is no evidence to recommend a specific scheme or systematic use of adjuvant chemotherapy. Most patients did not receive adjuvant chemotherapy. A combination of cyclophosphamide, cisplatin, doxorubicin and etoposide phosphate evidence found in case reports shows that 5FU-based chemotherapy, alone or in combination with cisplatin, is used. FU-based regimens are the most extended chemotherapy regimens used in gastrointestinal oncology with an acceptable toxicity profile (Table 6) [33,34,35].

## 7. ACC

### 7.1. Epidemiology, Clinical and Gross Features

Acinar cell carcinoma (ACC) of the pancreas is a rare and aggressive exocrine tumour that accounts for approximately 1% to 2% of all pancreatic malignancies [51]. It makes up roughly 12–15% of pancreatic cancers in adults and 15% in children. With a male/female ratio of 2:1, males are more typically affected. Adult patients’ mean age of diagnosis is 59 years (range 20–88 years) (Table 3) [52]. In rare cases, children can also be affected by ACC. Although ACC is most commonly found in the pancreatic head, jaundice is a rather uncommon sign [17]. When it comes to clinical features, the most common but non-specific symptoms are abdominal pain, weight loss, vomiting and nausea. A minority of patients (10–15%) with excessive lipase expression develop a classic lipase hypersecretion syndrome. It can present as fever, subcutaneous fat nodular necrosis, eosinophilia and polyarthralgia in patients with metastatic illness. This type of patient tends to have a worse prognosis [51].

At diagnosis, ACC’s average diameter varies between 8 and 10 cm. They are large, and usually solid; however, in rare cases, gross appearance can contain cystic changes. ACCs are well-circumscribed, usually encapsulated masses with a uniform, fleshy cut surface [17,53]. Haemorrhage, as well as necrosis regions, can be seen. Necrosis was found in 62% of ACCs, according to a prior study on the pathological examination of ACCs [54]. Breaking the continuity of the capsule and invasion of the tumour is a typical observation. Hence, the infiltration of the duodenum, large vasculature, stomach, kidney, peritoneum or spleen is a common finding, in about half of the patients [55]. ACC can also form a tumour plug within the main pancreatic duct [54].

Although most ACCs occur spontaneously, a few instances have been recorded in connection to Lynch syndrome and, very rarely, familial adenomatous polyposis (FAP) [52].

### 7.2. Diagnosis

Imaging is still one of the most effective and accurate tools for the early detection of tumours. Imaging of ACC can reveal significant cystic changes due to acinar secretion. Both MRI and CT are still very useful. CT is more effective in visualizing central calcifications. MRI is more sensitive to intratumoural haemorrhage or ductal dilatation and can distinguish between components of lesions and healthy tissues [56].

In establishing a definitive diagnosis of ACC, histopathological analysis is necessary. Immunohistochemistry is crucial to confirm the acinar cell differentiation (since the diagnostic hallmark of ACC is the demonstration of acinar differentiation, the demonstration of specific acinar cell products is often decisive). The antibodies used to identify acinar differentiation are those directed against trypsin, lipase, amylase and CEL. Trypsin, chymotrypsin and BCL-10 (clone 331.3, recognizing the COOH-terminal portion of carboxyl ester lipase) are the most sensitive and specific immunohistochemical markers recommended for diagnosis (Table 5) [17,52]. 

It is also important to mark that a recent study, which used large-scale tissue screening, underlines the excellent specificity of CPA1 (Carboxypeptidase A1) immunostaining for acinar differentiation in the pancreas. This study recommends incorporating CPA1 immunostaining in the diagnostic process of pancreatic tumours with ambiguous morphology [57].

Scattered chromogranin A and/or synaptophysin-positive neuroendocrine cells can be observed in several ACCs. Pancreatic ductal markers, such as cytokeratins 7 and 19, as well as hepatocellular markers, such as HepPar-1, glypican 3 and alpha-fetoprotein, may be expressed by ACC. Subgroups of ACCs strongly express p53, which can relate to a worse survival rate. However, further studies are needed to confirm the prognostic role of p53 expression [17,52].

The main differential diagnoses include pancreatoblastoma, pancreatic neuroendocrine neoplasm (PNETs), solid pseudopapillary tumours (SPTs) and pancreatic ductal adenocarcinoma [17]. The most common misdiagnoses are well-differentiated PNETs [51].

### 7.3. Genetics

In a recent study, whole-exome sequencing was performed on DNA extracted from eleven pure ACC surgical samples. Across four independent ACC studies, four consistently mutated genes were identified: FAT Atypical Cadherin 4 (FAT4), Mucin 5B (MUC5B), Titin (TTN), and Zinc Finger Homeobox 3 (ZFHX3). They were also able to identify three recurrent small deletions in over 30% of samples. This exploration of mutational signatures and mutation pathways led to the discovery of common mutations found in ACC. There is a possibility that targeting these affected pathways will improve clinical outcomes [58].

### 7.4. Management

It is difficult to exactly define the percentage of the 5-year survival of patients with ACC. There is still a lack of well-designed, large-scale prospective research, as the studies that have been published in recent years vary greatly in results. Age, stage IV, and the absence of surgery were all found to be independent indicators of poor overall survival [59].

It has been proven that comprehensive therapy based on radical resection, regardless of tumour size, is the best treatment option (Table 6). A study of 672 patients with acinar cell carcinoma of the pancreas showed that surgical resection notably improved survival. Patients with unresected ACC had a 22% 5-year survival rate, compared to 72% for those with resected ACC [53]. A 2020 study, which retrospectively reviewed 306 patients with confirmed ACC treated between January 2005 and August 2018, revealed that patients who had undergone resections had a 65.6% 5-year survival rate compared to 16.9% for those who had not. Patients in stage IV who received chemotherapy had a better prognosis than those who did not (median, 16 vs. 3 months) [59]. There is no consensus on adjuvant therapy for resected pancreatic ACC. Previous research into the effectiveness of adjuvant systemic therapy for acinar cell cancer had gathered mixed results. However, in a multivariate new study of 298 patients that took into account comorbid disease, lymph node and margin status, adjuvant systemic therapy provided after resection was linked to a higher overall survival rate than resection alone. Systemic therapy in the adjuvant setting can be beneficial, especially for patients who have pathologic evidence of lymph node involvement [60]. A few studies documented the use of 5-FU-based and gemcitabine-based chemotherapy. Compared to gemcitabine, oxaliplatin-based chemotherapy demonstrated better activity against pancreatic ACC. A 2017 study suggested that the use of modified FOLFIRINOX is safe and effective in the treatment of ACC (Table 6) [61].

Surgery is a possibly curative treatment for pancreatic acinar cell carcinoma that contributes to long-term survival and is recommended even in advanced diseases. Chemotherapy provides a chance to prolong the survival of metastatic patients.

## 8. Pancreatoblastoma

### 8.1. Epidemiology, Clinical and Gross Features

Pancreatoblastoma (PB) in children and young people is a malignant tumour. It is an extremely rare neoplasm that makes up nearly one percent of non-exocrine tumours of the pancreas in adults (about 70 cases of adult pancreatoblastoma are described in the literature); nevertheless, it remains the most common pancreatic tumour in children less than 10 years of age (Table 3), accounting in this case for 25% of all pancreatic tumours [20,62,63,64]. Its pathogenesis is mostly unknown; however, in some cases, pancreatoblastoma can be congenital-associated [1,65].

Prevalence is higher in the male population, especially in patients of Asian descent. The tumour occurs most often in the head of the pancreas; it is typically large and highly aggressive, showing multilineage differentiation, which is frequently acinar but may also be neuroendocrine, ductal or mesenchymal, suggesting primitive cellular origin [20,63,65]. Symptoms that most patients present include: abdominal mass or abdominal pain, abdominal distension and pyrexia [62].

### 8.2. Diagnosis

Differential diagnosis of PB is difficult due to its polyphenotypic nature. Clinical and laboratory investigations include serum alpha-fetoprotein (AFP), which may be elevated in up to a third of cases (similarly to hepatoblastoma); therefore, AFP is critical for diagnosis and monitoring the disease as a tumour marker [1]. Imaging studies used in diagnosing PBL include ultrasound, MRI and CT, which provide the most data about the lesion. Fine needle biopsy (using a percutaneous core needle, performed by laparotomy or laparoscopy) and immunohistochemical examination, which reveal markers for cytokeratin (CK) and alpha-1 antitrypsin (AAT), should be performed to confirm the diagnosis (Table 5). In contrast with SPN diagnosis, a biopsy is not recommended [62].

PBL is often mistaken with stroma-poor pancreatic neoplasms, but defining cytologic characteristics can distinguish it. In cytologic diagnosis, ancillary staining and β-catenin are used to highlight slight squamous morular areas. The crucial element in diagnosing PB is genetic testing; PB arises from germline mutations of the adenomatous polyposis coli gene (APC) [63]. In immunohistochemistry, trypsin, chymotrypsin, lipase and BCL10 are positive. PBs can focally express neuroendocrine markers, so it is important to differentiate from PanNENs by the expression of the above-mentioned markers and other specific features, such as squamous nests, predominant acinar differentiation and PASD-positive cytoplasmic granules. The PBL common genetic mutation is a loss of heterozygosity on chromosome 11p [64]. Molecular aberrations in the fibroblast growth factor receptor (FGFR) signalling pathway were described, concerning FGFR1 mutation, *FGFR2* gene rearrangement and a high mRNA expression of FGFRs 1, 3 and 4, as well as of their ligands, FGF3 and FGF4. [66]. Mutations in the *KRAS*, *TP53* and *CDKN2A/p16* genes are typically lacking in PBL, which can suggest that these neoplasms are genetically distinct from PDACs. Loss of SMAD4/DPC4 expression is rare in PB [64]. Additionally, associations of Beckwith-Wiedemann syndrome and familial adenomatous polyposis with PB have been mentioned in the literature [20].

Accurate differential diagnosis of PB, which is malignant and prone to recurrence, in preoperative imaging examination is very important for evaluation. Factors such as age, the size and border of the tumour, calcification, intratumoural haemorrhage and vessels, vascular invasion, the presence of distant metastasis and elevation of AFP differ significantly between these two types of pancreatic neoplasms [67].

The staging of PB follows the TNM classification of carcinoma of the exocrine pancreas [64].

### 8.3. Management

Complete surgical resection is the treatment of choice, with resection of hepatic metastases considered (Table 6) [1,62]. Pancreatic insufficiency and relapses seem more frequently described after a pylorus-sparing pancreatoduodenectomy (Traverso-Longmire) than with a classic Whipple surgery (14.3% vs. 5.7%) [32]. According to the strategy for the treatment of paediatric PB created by the European Cooperative Study Group for Paediatric Rare Tumors (EXPeRT), the first line of chemotherapy is a PLADO (cisplatin and doxorubicin) regimen (Table 6), classically used for hepatoblastoma. Neoadjuvant and postoperative PLADO chemotherapy (respectively, in unresectable cases or after incomplete resection) is performed as well. A total of 73% of patients with initially unresectable PBL had a tumour response to primary PLADO chemotherapy. After complete resection, postoperative PLADO chemotherapy administration is optional [36]. Radiation is potentially beneficial for palliation for patients with certain metastases, relapse or incomplete resection; however, in young children, potential side effects must be considered [35,53]. After the surgery, patients are monitored with ultrasound scans and serum AFP levels. Complete removal combined with chemotherapy is associated with long-term survival [62]. Contrarily, a very poor prognosis is given for paediatric patients with relapsed or unresectable cancer. Moreover, PB in adults prognosticates much worse than in children. Approximately 59% of patients develop metastases, mostly to the liver, lymph nodes or lungs. [64] The main goal for future treatments should be a better understanding of its oncogenic pathogenesis, which gives a chance for treatment in those groups where the prognosis is rather poor [20]. Follow-up is mandatory due to the risk of relapse, but there is no protocol yet established. Screening and genetic consultations should be offered to patients with PB and their families. Moreover, emphasis should be put on children with genetic syndromes for the development of other embryonal tumours (Beckwith–Wiedemann syndrome) or gastrointestinal tumours (FAP) [36].

## 9. Other Rare Non-Neuroendocrine Pancreatic Tumours

### 9.1. Pancreatic Sarcomas 

Most of the cases are secondary to retroperitoneal sarcoma. It mostly affects adults 50 years old and above. Leiomyosarcoma, the most common primary sarcoma, is associated with a poor prognosis. It is often metastatic at the time of presentation. Surgical resection of the localised tumour is the treatment of choice. Leiomyomas in the pancreas are very rare. Regardless of their benign character, these tumours are usually treated as malignant [1].

### 9.2. Pancreatic Squamous Cell Carcinoma

Adenosquamous carcinoma of the pancreas (ASCAP) constitutes about two percent (1–4%) of all pancreatic non-endocrine lesions. It presents similarly to adenocarcinoma of the pancreas, but has a worse overall prognosis, with a survival rate less than 2 years. Surgical resection remains the best therapeutic option for patients with ASCAP, giving curable potential with adjuvant or neoadjuvant therapy depending on clinical and pathological staging. Further research to determine specific tumour genetic markers is necessary to adjust the treatment [68].

## 10. Conclusions

Although the most common exocrine pancreatic tumour, PDAC, has the highest malignant potential, it is still essential to properly classify and differentiate less-common lesions. We intended to organise current information about six of these infrequent tumours and highlight discoveries regarding each neoplasm with a focus on differentiation, management and risk of malignancy. It is vital to continue the search for new biomarkers, genetic mutations and the development of more specific biochemical tests for determining malignancy in rare pancreatic neoplasms, especially in cystic lesions. Frequent guideline updates can help to avoid misdiagnosis, which could lead to unnecessary resections or oversight of malignant transformation.

## Figures and Tables

**Figure 1 cancers-15-02216-f001:**
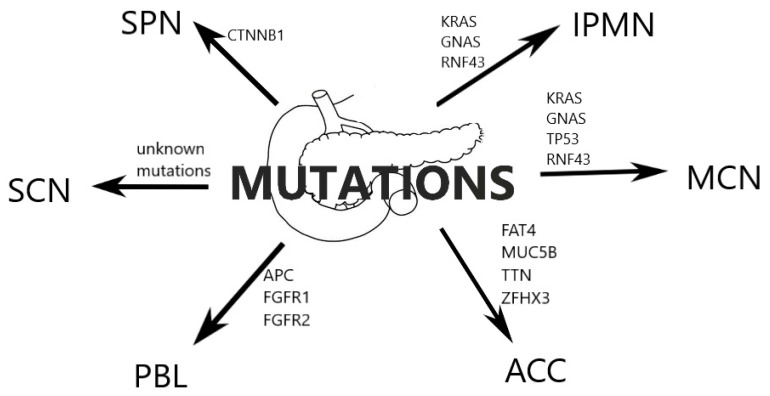
Most common mutations associated with benign and malignant pancreatic tumours.

**Table 1 cancers-15-02216-t001:** WHO classification of tumours of the pancreas with corresponding ICD-O codes [4].

Benign	Serous cystadenoma, NOS (8441/0)
	Intraductal papillary mucinous neoplasm - with low-grade intraepithelial neoplasia (8453/0) - with high-grade intraepithelial neoplasia (8453/2) - with associated invasive carcinoma (8453/3)
	Mucinous cystic neoplasm - with low-grade intraepithelial neoplasia (8470/0) - with high-grade intraepithelial neoplasia (8470/2) - with associated invasive carcinoma (8470/3)
Malignant	Acinar cell carcinoma (8550/3)
	Acinar cell cystadenocarcinoma (8551/3)
	Mixed acinar-neuroendocrine carcinoma (8154/3)
	Mixed acinar-endocrine-ductal carcinoma (8154/3)
	Mixed acinar-ductal carcinoma (8552/3)
	Pancreatoblastoma (8971/3)
	Solid pseudopapillary neoplasm of the pancreas (8452/3) - Solid pseudopapillary neoplasm with high-grade dysplasia

**Table 2 cancers-15-02216-t002:** Classification of IPMNs based on cytological features and immunochemistry [2].

Type	Mucins
Gastric	MUC5AC+ MUC6+
Intestinal	MUC2+ CDX2+
Pancreato-biliary	MUC1 strong

**Table 3 cancers-15-02216-t003:** Comparison of clinical features, management and prognosis of rare pancreatic tumours.

	IPMN	MCN	SCN	ACC	SPN	Pancreatoblastoma
Incidence	3–5%	1–2%	1–2%	1%	1–2%	<1%
M/F ratio	1:1	1:20	1:2	2:1	1:10	2:1
Malignancy potential	MD 50% BD 15%	4.4–16.6%	0%	100%	15%	High
Median age	69	48	61	59	28	5
Location	Head	Body/tail	Body/tail	Head/body/tail	Tail	Head
Mean size [mm]	MD 28.8 BD 29.1	20–250	46.1	80–100	20–150	20–200
Cystic/Solid	Cystic	Cystic	Cystic	Solid (rarely cystic)	Solid	Solid
5-year survival	MD 83% BD 88%	26–75% (invasive MCN)	97%	25–50%	97%	95%
Biomarkers	CEA > 192 ng/mL High amylase	CEA > 192 ng/mL Low amylase	CEA < 192 ng/mL VEGF-A > 5000 pg/mL	Trypsin Chymotrypsin BCL-10	CTNNB1 mutations, β-catenin, E-cadherin	High AFP CK AAT
Histological features	Tall papillary structures lined by epithelium Mucin production	Ovarian cortical cells Mucin production	Low cuboidal cells with clear cytoplasm Sponge-like structure in microcystic type	Acinar cell differentiation	Sheets of epithelioid cells with scant cytoplasm Pseudopapillary structures	Mixed epithelial and mesenchymal components Vascular and perineural elements

**Table 4 cancers-15-02216-t004:** Comparison of selected fluid biomarker characteristics of pancreatic cystic neoplasms—differential diagnosis.

	IPMN	MCN	SCN
Biomarkers			
CEA	>192 ng/mL	>192 ng/mL	<192 ng/mL
CA19-9	>50,000 U/mL	>50,000 U/mL	-
VEGF-A	-	-	>5000 pg/mL
Amylase	High	Low	Low
Glucose	<50 mg/dL	<50 mg/dL	>50 mg/dL
Mutations	KRAS, GNAS	KRAS, GNAS, TP53, RNF43	-
Histological Features	Papillary structures, atypia	Ovarian-like columnar cells, mucus production	Simple cuboidal epithelial cells
Cyst Fluid Viscosity	High	High	Low

**Table 5 cancers-15-02216-t005:** Comparison of selected characteristics of diagnostic markers found in rare pancreatic tumours [14,16,17,18,19,20].

Marker	Diagnosis	Cut-Off Value	Material	Sn/Sp (%)	Tumours	Notes
CEA	Malignant	>5 µg/L	Serum	40/92	IPMN, MCN	80% of patients with invasive IPMN had increased levels of CEA
	Cystic	>192 ng/mL	Cystic fluid	73/84	IPMN, MCN	
		≤10 ng/ml	Cystic fluid	96/82	SCN	
CA19-9	Malignant	>37 U/mL	Serum	74/86		
	Mucinous	>50,000 U/mL	Cystic fluid	75/90	IPMN, MCN	
CEA/ CA19-9	Malignant	>5 µg/L/>37 U/mL	Serum	80/82	IPMN, MCN	Accuracy of 81%
VEGF-A	Serous	>5000 pg/mL	Cystic fluid	100/84	SCN	Elevated levels of VEGF-A may also occur in PNETs
VEGF-A/CEA	Serous	>5000 pg/mL/≤10 ng/mL	Cystic fluid	96/100	SCN	
VEGF-C	Serous	>200 pg/mL	Cystic fluid	100/90	SCN	
MUC6	Serous	NA	Cystic fluid	100/100	SCN	PNET differentiation
Calponin	Serous	NA	Cystic fluid	71/100	SCN	PNET differentiation
Glucose	Mucinous	<66 mg/dL	Cystic fluid	94/64	MCN	
Amylase	Pseudocyst	>250 U/mL	Cystic fluid	44/98	-	High amylase levels may be associated with IPMN
IMP3	Malignant	NA	Cystic fluid	78/96	IPMN, MCN	
mAb Das-1	Malignant	NA	Cystic fluid	88/99	IPMN	
KRAS	Malignant	NA	Cystic fluid	45/96	IPMN, MCN	
KRAS/GNAS	Malignant	NA	Cystic fluid	65/100	IPMN, MCN	
Trypsin	Acinar cell	NA	Tissue	NA	ACC	Negative in SPN
BCL-10	Acinar cell	NA	Tissue	NA	ACC	Negative in SPN
β-catenin/ E-cadherin	SPN	NA	Tissue	NA	-	β-catenin is expressed in over 90% of SPN
CD10	SPN	NA	Tissue	NA	-	
Cytokeratin	Pancreatoblastoma	NA	Tissue	NA	-	
AAT	Pancreatoblastoma	NA	Tissue	NA	-	

Sn—sensitivity, Sp—specificity, NA—statistically significant but Sn/Sp not reported, CA19-9—cancer antigen 19-9, CEA—carcinoembryonic antigen, MUC—mucin, KRAS—Kirsten rat sarcoma viral oncogene homolog, GNAS—guanine nucleotide-binding protein.

**Table 6 cancers-15-02216-t006:** Summarised treatment of different tumours.

	IPMN	MCN	SCN	ACC	SPN	Pancreatoblastoma
Primary/preferred method	Resection (with adjuvant therapy if invasive)	Resection (>40 mm/ symptomatic/ have risk factors) or surveillance [9]	Surveillance	Resection (regardless of tumour size) [29]	Resection [30]	Resection (with resection of hepatic metastases considered) [1]
Surgical resection	Pancreatoctomy (with lymph node dissection for invasive IPMN) [31]	Distal pancreatico-splenectomy with or without lymph node dissection [23]	Not recommended	Radical resection [29]	Classic Whipple surgery [32]	Classic Whipple surgery [32]
Adjuvant therapy	FOLFIRINOX or gemcitabine [25]	gemcitabine and 5-fluorouracil [9]	NA	FOLFIRINOX [33]	5FU-based chemotherapy alone or in combination with cisplatin [33,34,35]	PLADO (cisplatin and doxorubicin) [36]

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
