# Peer review of "Rare Non-Neuroendocrine Pancreatic Tumours"

_cancers, 2023, doi:10.3390/cancers15082216_

Round 1
Reviewer 1 Report
The author gathered the latest data about the epidemiology, diagnosis, biomarkers, and management of six rare pancreatic tumours: intraductal papillary mucinous neoplasm, mucinous cystadenoma, serous cystic neoplasm, acinar cell carcinoma, solid pseudopapillary neoplasm and pancreatoblastoma.
Specific suggestions:
1. It is more appropriate to change the title to "Rare non-neuroendocrine Pancreatic tuner”.
2. The classification section: it is recommended to change the description of benign and malignant pancreas tumorsto a tabular format
3. IPMN section: there was no “Table 1” in the main body; IPMN related survival data were less discussed, and it is recommended to describe recent published data; it is recommended to describe the detail follow-up plan for IPMN patients with low risk of malignancy; postoperative adjuvant therapy for malignant IPMN should be disscussed.
4. It is recommended to describe the adjuvant treatment and postoperative adjuvant treatment plans for malignant MCN.
5. It is recommended to provide typical imaging images, tumor body images, and HE images of each type of tumor.
6. It is recommended to summarize the treatment of different tumors and present them in tabular form.
7. It is recommended to add "Other Rare non neuroendocrine pancreatic tumors" section to briefly describe other pathological types of rare pancreatic tumors, such as pancreatic metastatic tumor, pancreatic sarcomas, and pancreatic squamous cell carcinoma
Reviewer 2 Report
This is a well-written manuscript titled “Rare pancreatic tumours” that discuss the latest data about the epidemiology, diagnosis, biomarkers, and management of six rare pancreatic tumours. The manuscript described six rare pancreatic tumours – intraductal papillary mucinous neoplasm (IPMN), mucinous cystadenoma (MCN), serous cystic neoplasm (SCN), acinar cell carcinoma (ACC), solid pseudopapillary neoplasm (SPN) and pancreatoblastoma (PB) and covers every detail about the newest reports on the courses of treatment and systematized differential diagnosis. The manuscript is clear, comprehensive and relevant for readers to understand rare pancreatic tumours. The manuscript can be accepted with minor revisions.
1. Author should include some figures to explain the molecular mechanism of benign and malignant pancreatic tumour formation.
2. Author should explain the clinical and gross features of all these six rare pancreatic tumours with some scientific illustration.
Reviewer 3 Report
The review article entitled: "Rare Pancreatic Tumours"; by Agata Mormul et al., is overall well written, introduced and structured. The content covers perfectly the topic; however, it needs some amendments to improve the quality and scientific soundness before publication.
Please find below my comments:
-Concerning IPMN:
In my consideration, it is important to emphasize that IPMN apparently progress from low-grade dysplasia (clinically benign) to high-grade dysplasia, and that neoplasms with high-grade dysplasia sometimes become invasive. This information comes from AFIP (Armed Foces Institute of Pathology).
Ref: Hruban RH, Pitman MB, Klimstra DS. Tumors of the pancreas. In: Atlas of tumor pathology, 6th ed, Armed Forces Institute of Pathology, Washington, DC 2007. Vol 4.
About the management of IPMN, it may be appropriate to note:
1) Surveillance could be an option in older patients, cysts lacking high-risk features or who are unfit for surgery.
2) In patients who invasive adenocarcinoma is demonstrated after surgical resection adjuvant chemotherapy should be always considered according to performance status (PS). FOLFIRINOX for younger patients with PS 0-1 or gemcitabine based regimens for order patients and/or PS ≥2.
References:
Number 9º of manuscript: European evidence-based guidelines on pancreatic cystic neoplasms.
DOI: 10.1056/NEJMoa1809775
- Concerning Mucinous cystoadenoma
Ovarian type stroma is considered classically a key diagnostic feature for MC diagnosis after resection, it helps to differentiate if from IPMN. Due to its relevance, I consider it could be helpfull to add in the same paragraph some lines about:
1) Hypothesis about possible relationship between stroma and MC pathogenesis
2)Whether ovarian-like stroma is a mere morphologic resemblance or there is a functional similarity to the true ovarian stroma.
References:
DOI: 10.1097/00129039-200406000-00009
DOI: 10.1309/U2BBP4EMBAHCM6E6
-Concerning Serous cystic neoplasm:
It could be helpful to add some classical histopathological features that are important for diagnosis, as it is done in previous points. In the beginnig only cuboidal epithelium is mentioned. In my opinion, I would add:
1)The majority of serous cystadenomas are composed of multiple small cysts, and they have a classic "honeycomb appearance." Histochemically, the cytoplasm stains positive with the periodic acid-Schiff (PAS) stain due to the presence of glycogen. Also, multiple serous cystadenomas could be found in patients with von Hippel-Lindau syndrome.
-Concerning Solid pseudopapillary neoplasm:
I would add that chemotherapy have been used in neoadjuvant and postoperative setting however there is no evidence to recommend a specific scheme or systematic use of adjuvant chemotherapy. Most patients did not receive adjuvant chemotherapy .
Apart from the combination of cyclophosphamide, cisplatin, doxorubicin and etoposide phosphate evidence found in case reports shows that 5FU-based chemotherapy alone or in combination with cisplatin is used. FU-based regimens are the most extended chemotherapy regimens used in gastrointestinal oncology with an acceptable toxicity profile.
References:
DOI: 10.1111/ans.15701
DOI: 10.1016/j.amsu.2021.102708
DOI: 10.3389/fped.2022.899965
- There is no comments for review from pancreatoblastoma and acinar cell carcinoma.
Round 2
Reviewer 1 Report
None
Reviewer 3 Report
Thank for providing the amended version of the manuscript.